# Survival outcomes in elderly Taiwanese women according to breast cancer subtype and lymph node status: A single-center retrospective study

**Kung-Hung Lin[1,2], Huan-Ming Hsu[1], Kuo-Feng Hsu[1], Chi-Hong Chu[1], Zhi-Jie Hong[1], Chun-Yu Fu[1], Yu-Ching Chou[3], Golshan Mehra[4], Ming-Shen Dai[5], Jyh-Cherng Yu[1], Guo-Shiou Liao[1] ***

1 Division of General Surgery, Department of Surgery, Tri-Services General Hospital, National Defense Medical Center, Taipei, Taiwan, 2 Division of General Surgery, Department of Surgery, Zuoying Branch of Kaohsiung Armed Forces General Hospital, Kaohsiung, Taiwan, 3 School of Public Health, National Defense Medical Center, Taipei, Taiwan, 4 Department of Surgery, Brigham and Women's Hospital and Dana Farber Cancer Institute, Boston, Massachusetts, United States of America, 5 Division of Hematology/Oncology, Tri-Service General Hospital, National Defense Medical Center, Taipei, Taiwan

* guoshiou@ndmctsgh.edu.tw

**Data Availability Statement:** Data cannot be shared publicly because the study's retrospective protocol was reviewed and approved by the Tri-

## Abstract

This study aimed to determine the rates of overall survival and recurrence-free survival among elderly Taiwanese women (>65 years old) according to breast cancer subtype and lymph node status. We identified 554 eligible patients who were >65 years old and had been treated based on international recommendations at our center between June 2005 and June 2015. Patients with the luminal A subtype had the highest rates of overall survival (90.6%) and recurrence-free survival (97.0%), while the lowest overall survival rate was observed in those with the triple-negative subtype (81.3%) and the lowest recurrence-free survival rate was observed in those with the luminal B subtype (84.0%). Multivariate Cox proportional hazard analysis, using the luminal A subtype as the reference, revealed significant differences in recurrence-free survival among luminal B patients according to lymph node status. Among elderly Taiwanese women with breast cancer, the breast cancer subtype might help predict survival outcomes. The luminal B subtype was associated with poor recurrence-free survival, and lymph node status was useful for predicting recurrence-free survival in this subset of patients.

## Introduction

The elderly population is increasing worldwide, and the proportion of Taiwanese women who are ≥65 years old is expected to increase from 14% in 2018 to 20% in 2025 [1]. Breast cancer is the most common cancer among women and the leading cause of cancer-related deaths worldwide [2]. The Taiwanese National Health Insurance database was used to estimate the annual

Service General Hospital's human investigations committee (1-107-05-135). Data are available from the Tri-Service General Hospital's human investigations committee Institutional Data Access / Ethics Committee (via Miss Chen, email-1: irbtzu@ndmctsgh.edu.tw, email-2: help@cims.tw, phone-1:+886-2-8792-3311#10675, phone-2: +886-2-8792-3311#10807) for researchers who meet the criteria for access to confidential data.

**Funding:** The authors received no specific funding for this work.

**Competing interests:** The authors have declared that no competing interests exist.

prevalence and incidence of breast cancer between 1997 and 2013, which revealed a prevalence of 834.37 per 100,000 persons and an incidence of 93.00 per 100,000 persons in 2013.

In Taiwan, the standardized incidence was 52.34 per 100,000 person-year in 1997 and 93.00 per 100,000 person-year in 2013 [1, 3]. The age-standardized incidence rates (ASIR) have gradually increased over the past several years, with an incremental annual change of 3.5 per 100,000 persons. It suggests that the breast cancer incidence increased over the study period [2, 3]. Moreover, the fastest-growing population segment includes individuals who are ≥65 years old, and breast cancer is relatively common among women in this age group.

Recent advances in molecular testing have allowed breast cancers to be categorized into clinically relevant molecular subtypes. According to the St. Gallen International Breast Cancer Conference (2011), breast cancer subtypes (BCSs) are classified as luminal A, luminal B, luminal human epidermal growth factor receptor 2 (HER2), HER2, and triple negative (TN) [4–6]. Lymph node (LN) status is another factor that is strongly related to overall survival (OS) in breast cancer patients and has been an integral component in the staging, prognostication, and treatment of invasive breast cancers [6, 7]. Furthermore, recent data have indicated that the prognosis varies according to BCS [8]. However, the survival rates of elderly patients with breast cancer according to BCS and LN status have not been fully examined. Therefore, in this retrospective study, we evaluate and report the rates of OS and recurrence-free survival (RFS) among elderly Taiwanese breast cancer patients according to their BCS, as well as according to their LN status. To the best of our knowledge, this study includes the largest series of patients reported in the literature to date.

## Materials and methods

### Database

This retrospective study evaluated data from the electronic medical records of patients who were diagnosed with invasive breast cancer and underwent potentially curative surgery at the Tri-Service General Hospital (TSGH, Taipei, Taiwan) between June 2005 and June 2015. The information recorded for each patient included the age at diagnosis, year of diagnosis, and date of death or last contact. All patients had undergone either mastectomy or breast-conserving surgery, with subsequent endocrine therapy, local radiotherapy, or adjuvant systemic treatment, selected based on international recommendations [8–12]. We obtained follow-up data from the clinical history or over the phone. For deceased patients, the date and cause of death were also collected. The total incidences of recurrence or death due to breast cancer were determined based on follow-up visits that were conducted until October 2016. Living patients or patients without follow-up were censored at the end of the follow-up period. In our organization, all patients will be followed up for the first five years for six months and then once a year for the next five years. Follow-up examinations include physical examinations and blood tests to evaluate tumor markers, and it is recommended to have a bilateral mammogram (after lumpectomy) or the remaining contralateral breast (after mastectomy) for a follow-up examination every year. CT and MRI scans were not our routine follow-up inspection items. Tumor characteristics included tumor size (≤ 2, 2 to 5, and >5 cm); tumor pathologic stage (I, II, III, IV); status of ER, PR, and HER2 (positive, negative, or unknown); and LN status (negative or positive). Treatment factors included radiotherapy, type of surgery, chemotherapy, or endocrine therapy. The tumor pathologic stage was defined by the tumor node metastasis (TNM) classification as proposed by the American Joint Committee on Cancer (AJCC) for grouping patients with respect to prognosis.

## Selection of the study subjects

According to the American Joint Cancer Council (AJCC) standards, the breast cancer database includes 4,363 newly diagnosed stage I, II, III, or IV patients in the Tri-Service General Hospital from 2005 to 2015. Exclude breast cancer patients< 65 years of age (n = 3,809). The exclusion conditions are as follow patients with other cancer diagnoses before or after the initial breast cancer diagnosis, patients lacking correct ER, PR, HER2 data or missing data, diseases with unknown surgical information or missing surgical data after the initial breast cancer diagnosis, lack of relevant information patients with tumor size or the number of positive axillary lymph nodes and patients who have never had a disease or whose recurrence date is missing or wrong. Finally, a total of 503 patients were eligible for analysis between age and clinicopathological characteristics including 203 luminal A breast cancer, 128 luminal B breast cancer, 88 luminal B2 breast cancer, 20 HER2+ breast cancer, and 32 TN breast cancer (Fig 1).

The hospital started to promote the use of electronic medical records in 2001. The cancer center was formally established in 2002. Since then, the data of cancer patients began to be digitalized and registered year by year. At this time, electronic diseases are managed by using barcodes and radio frequency identification technology (radio frequency identification; RFID) to attach to the paper medical records, and then use database index and other technologies to record the location and flow of the paper medical records to achieve digitalization. From 2002 to 2009, our hospital adopted a mixed medical record model (paper medical records and electronic medical records exist at the same time). Because the legal status of electronic medical records has not been established and part of the electronic medical record system is not yet complete, our hospital adopts a transitional mixed medical record model. Between 2005 and 2009, we adopted large-scale manual methods to transfer paper medical records into electronic transcripts. At the same time, the construction of the electronic database of cancer patients in our hospital was completed from 2009 to 2010. Since then, the medical records of cancer patients have been all electronic.

The study's retrospective protocol was reviewed and approved by the Tri-Service General Hospital's human investigations committee (1-107-05-135). The requirement for informed consent was waived due to the retrospective nature of the study.

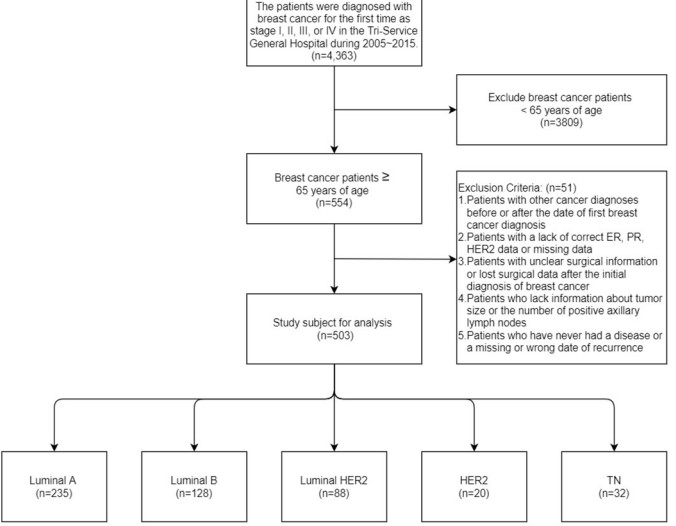

**Fig 1. Flowchart presenting the process of selecting the study subjects.**

## Tumor characteristics

Tumor-related data included those of LN status (negative or positive), tumor size (<2 cm, 2–5 cm, or >5 cm), and BCS. If data on the Ki-67 index were not available, some alternative measure of proliferation, such as the histological grade, was used to identify the BCS, as previously reported [6, 8, 13]. Intrinsic subtypes were classified into five groups based on immunohistochemistry findings: luminal A (estrogen receptor-positive [ER+] and/or progesterone receptor-positive [PR+], HER2−, and histological grade 1 or 2), luminal B (ER+ and/or PR+, HER2−, and histological grade 3), luminal B2 (ER+ and/or PR+, HER2+, and any histological grade), HER2+ (ER−, PR−, and HER2+), and TN (ER−, PR−, and HER2−). Immunohistochemistry was used to identify positively stained nuclei for determining ER/PR positivity (>1%). Tumors were considered HER2+ when the cells exhibited strong membrane staining (3+), and tumors were considered HER2− when they exhibited 0 or 1+ staining for HER2 protein expression. For patients with an equivocal membrane staining score for HER2 (2+), fluorescence in situ hybridization was performed to evaluate gene amplification [12, 14].

## Statistical analysis

Continuous data are expressed as mean ± standard deviation, and categorical data are expressed as number (percentage). Survival intervals were calculated from the date of cancer diagnosis to the date of death because of any cause or the last follow-up (OS) or from the date of cancer diagnosis to the date of the first detected relapse or last follow-up without relapse (RFS). The chi-squared test was used to analyze categorical clinicopathological variables, and differences in OS and RFS according to BCS were analyzed using the Kaplan–Meier method and log-rank test. Multivariate Cox proportional hazard analysis was used to calculate adjusted mortality risks (hazard ratio [HR] and 95% confidence interval [CI]) and identify factors that best predicted OS and RFS. Differences were considered statistically significant at two-sided P-values <0.05. All statistical analyses were performed using IBM SPSS software (version 22.0; IBM Corp., Armonk, NY, USA).

## Results

### Clinicopathological characteristics

The distribution of BCSs among the 554 elderly Taiwanese women (>65 years old) was as follows: luminal A (46.7% of patients), luminal B (25.4% of patients), luminal B2 (17.5% of patients), HER2 (4.0% of patients), and TN (6.4% of patients) (Table 1). Significant differences were observed according to BCS in terms of tumor size (P < 0.001) and LN status (P = 0.001). Relative to the other subtypes, the luminal A subtype was the most likely to involve a smaller tumor size (49.4%) and the least likely to show LN involvement (31.5%). The HER2 subtype had the highest incidence of LN involvement (45.0%), while the TN subtype was the most likely to involve a large tumor size (28.1%). Most patients (88.9%) underwent surgery, which included breast-conserving surgery (12.3%) or modified radical mastectomy (76.6%). Other treatments included radiotherapy (24.4%), adjuvant chemotherapy (81.7%), and endocrine treatment (76.8%).

### Survival outcomes

Patients with the luminal A subtype had the highest 10-year rates of OS (90.6%) and RFS (97.0%). The lowest OS rate was observed for those with the TN subtype (81.3%), and the

**Table 1. Clinicopathological characteristics of all patients according to breast cancer subtype (age > 65, n = 503).**

| Variable | Luminal A (%) | Luminal B (%) | Luminal B2 (%) | HER2 (%) | TN (%) | P-value |
|---|---|---|---|---|---|---|
| Number of cases | 235 (46.7) | 128 (25.4) | 88 (17.5) | 20 (4.0) | 32 (6.4) | |
| Tumor size | | | | | | <0.001* |
| ≤2 cm | 116 (49.4) | 39 (30.5) | 25 (28.4) | 9 (45.0) | 7 (21.9) | |
| >2–5 cm | 95 (40.4) | 68 (53.1) | 55 (62.5) | 10 (50.0) | 9 (28.1) | |
| >5 cm | 24 (10.2) | 21 (16.4) | 8 (9.1) | 1 (5.0) | 16 (50.0) | |
| Lymph node status | | | | | | 0.001* |
| Negative | 158 (79.4) | 64 (61.0) | 48 (62.3) | 11 (55.0) | 18 (64.3) | |
| Positive | 41 (31.5) | 41 (39.0) | 29 (37.7) | 9 (45.0) | 10 (35.7) | |
| Operation type | | | | | | 0.335 |
| No | 27 (11.8) | 17 (13.7) | 7 (8.3) | 0 (0.0) | 3 (10.0) | |
| Breast conservation surgery | 30 (13.1) | 12 (9.7) | 10 (11.9) | 1 (5.0) | 7 (23.3) | |
| Modified radical mastectomy | 172 (75.1) | 95 (76.6) | 67 (79.8) | 19 (95.0) | 20 (66.7) | |
| Radiotherapy | | | | | | 0.008* |
| No | 188 (82.1) | 87 (70.2) | 63 (74.1) | 14 (70.0) | 17 (56.7) | |
| Yes | 41 (17.9) | 37 (29.8) | 22 (25.9) | 6 (30.0) | 13 (43.3) | |
| Chemotherapy | | | | | | 0.008* |
| No | 40 (18.6) | 14 (11.8) | 13 (16.7) | 8 (40.0) | 9 (34.6) | |
| Yes | 175 (81.4) | 105 (88.2) | 65 (83.3) | 12 (60.0) | 17 (65.4) | |
| Endocrine therapy | | | | | | <0.001* |
| No | 25 (11.2) | 15 (12.4) | 23 (27.7) | 20 (100.0) | 28 (93.3) | |
| Yes | 199 (88.8) | 106 (87.6) | 60 (72.3) | 0 (0.0) | 2 (6.7) | |
| Overall survival | | | | | | 0.185 |
| Deceased | 22 (9.4) | 22 (17.2) | 11 (12.5) | 2 (10.0) | 6 (18.8) | |
| Alive | 213 (90.6) | 106 (82.8) | 77 (87.5) | 18 (90.0) | 26 (81.3) | |
| Recurrence-free survival | | | | | | 0.002* |
| Deceased | 6 (3.0) | 17 (16.0) | 6 (8.1) | 1 (6.3) | 2 (6.7) | |
| Alive | 195 (97.0) | 89 (84.0) | 68 (91.9) | 15 (93.7) | 28 (93.3) | |

HER2, human epidermal growth factor receptor 2; TN, triple negative.

*Significant at P < 0.05.

lowest RFS rate was observed for those with the luminal B subtype (84.0%) (Table 1). The Kaplan–Meier curves for OS and RFS according to the BCS are shown in Figs 2 and 3.

The difference in OS according to the BCS was not statistically significant (P = 0.135), although there was a significant difference in RFS according to the BCS (P = 0.002). In the multivariate Cox proportional hazard analysis, age of >65 years was independently associated with poor OS and RFS in the BCS subgroups after controlling for tumor size, LN status, radiotherapy, surgery type, chemotherapy, and hormone therapy (Table 2). Therefore, the luminal A subtype was used as the reference group, and we found that patients with the luminal B subtype had significantly poorer RFS (HR: 4.076, 95% CI: 1.426–11.649; P = 0.009), even after adjusting for tumor size, LN status, radiotherapy, surgery type, chemotherapy, and hormone therapy (Table 2). Furthermore, many investigators have reported a statistically significant association between the BCS and LN metastasis [6, 8, 9, 15, 16]. Therefore, we evaluated the OS and RFS according to the BCS and LN status (Table 3), which revealed a significant difference in RFS according to LN status among patients with the luminal B subtype (HR: 14.427, 95% CI: 1.409–147.740; P = 0.025).

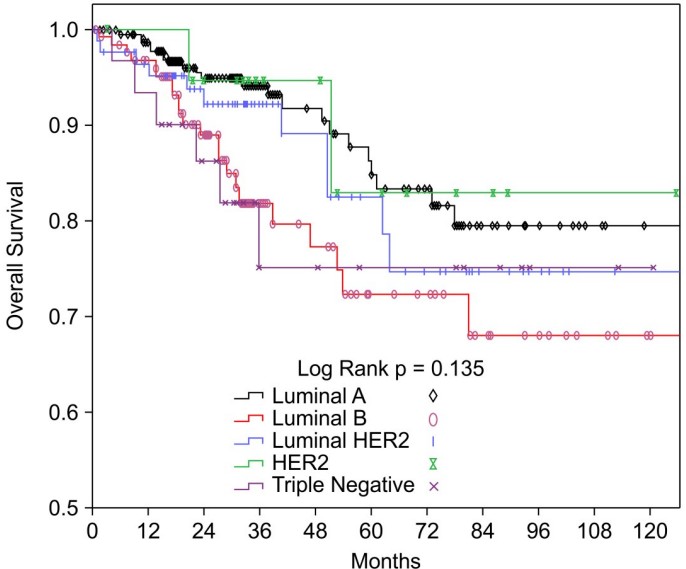

**Fig 2. Kaplan–Meier curves of overall survival according to lymph node status and breast cancer subtype.** HER2, human epidermal growth factor receptor 2.

## Discussion

Breast cancer is the second most common cancer worldwide and the most common cancer among women, with an estimated 1.67 million new cases diagnosed in 2012 (25% of all cancers) [2]. Furthermore, the elderly population is increasing worldwide, and breast cancer in elderly women is a major challenge for modern healthcare systems. Previous studies have indicated that advanced age is associated with more favorable tumor biology and that breast

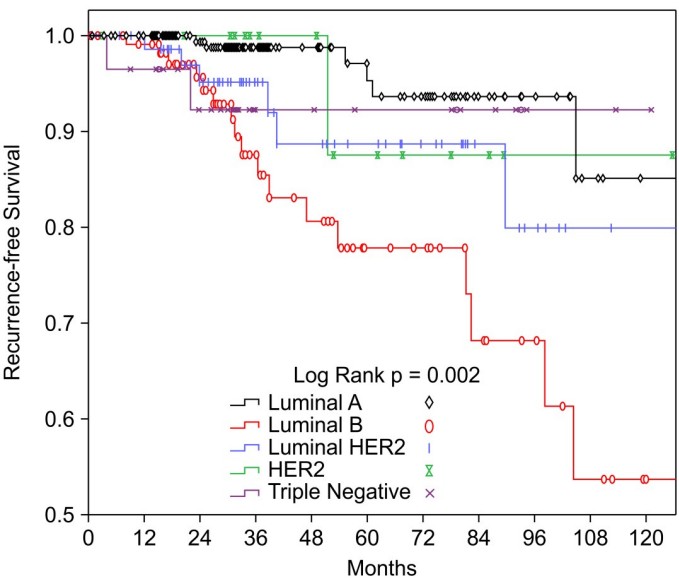

**Fig 3. Kaplan–Meier curves of recurrence-free survival according to lymph node status and breast cancer subtype.** HER2, human epidermal growth factor receptor 2.

**Table 2. Multivariate analysis of overall and recurrence-free survival according to breast cancer subtype.**

|  | Luminal A | Luminal B | Luminal B2 | HER2 | TN |
|---|---|---|---|---|---|
|  | P-value and HR (95% CI) |  |  |  |  |
| Overall survival |  |  |  |  |  |
|  | 1 (reference) | P = 0.127 1.935 (0.829–4.518) | P = 0.280 1.702 (0.648–4.469) | P = 0.874 1.165 (0.178–7.612) | P = 0.098 3.883 (0.777–19.398) |
| Recurrence-free survival |  |  |  |  |  |
|  | 1 (reference) | P = 0.009* 4.076 (1.426–11.649) | P = 0.299 2.028 (0.534–5.951) | P = 0.528 0.465 (0.043–5.024) | P = 0.09 2.257 (0.880–5.790) |

HER2, human epidermal growth factor receptor 2; TN, triple negative; HR, hazard ratio; CI, confidence interval.

The model was adjusted for tumor size, lymph node status, radiotherapy, surgery type, chemotherapy, and hormone therapy.

*Significant at P < 0.05.

cancer-related survival in elderly women is similar to that in the general population, regardless of disease status [17]. For example, Kim et al. [18] compared 4388 patients with invasive breast cancer according to age (<65 and ≥65 years) and reported a median age of 47 years (range: 18–91 years) and 317 patients (7.2%) who were ≥65 years old. Their results indicated that the tumor characteristics were similar between the two age groups. Other investigators have demonstrated that the TN subtype is associated with a large tumor size [19–22], and Liao et al. [8] have reported that the highest risk of LN metastasis was observed for the luminal B and luminal B2 subtypes. The present study also revealed that the TN subtype was associated with a larger tumor size and that LN positivity was the most common among elderly breast cancer patients with the HER2 subtype. Interestingly, patients with the HER2 subtype were most likely to undergo modified radical mastectomy (95%).

**Table 3. Survival outcomes according to lymph node status and breast cancer subtype.**

|  | Luminal A | Luminal B | Luminal B2 | HER2 | TN |
|---|---|---|---|---|---|
|  | P-value and HR (95% CI) |  |  |  |  |
| Overall survival |  |  |  |  |  |
| *Lymph node status* |  |  |  |  |  |
| Negative | 1 (reference) | P = 0.386 1.671 (0.523–5.338) | P = 0.765 0.786 (0.162–3.808) | P = 0.305 5.307 (0.219–128.530) | P = 0.985 0.000 (0.000–undefined) |
| Positive | 1 (reference) | P = 0.081 4.328 (0.834–22.471) | P = 0.062 5.148 (0.923–28.714) | P = 0.708 0.602 (0.042–8.613) | P = 0.083 6.009 (0.791–45.664) |
| Recurrence-free survival |  |  |  |  |  |
| *Lymph node status* |  |  |  |  |  |
| Negative | 1 (reference) | P = 0.098 3.101 (0.811–11.855) | P = 0.689 1.429 (0.248–8.231) | P = 0.992 0.000 (0.000–undefined) | P = 0.988 0.000 (0.000–undefined) |
| Positive | 1 (reference) | P = 0.025* 14.427 (1.409–147.740) | P = 0.097 9.470 (0.668–134.319) | P = 0.805 0.694 (0.038–12.578) | P = 0.892 1.227 (0.063–23.762) |

HER2, human epidermal growth factor receptor 2; TN, triple negative; HR, hazard ratio; CI, confidence interval.

The model was adjusted for tumor size, radiotherapy, surgery type, chemotherapy, and hormone therapy.

*Significant at P < 0.05.

While there was no significant difference in OS according to the BCS, we did detect a significant difference in RFS according to the BCS (luminal B vs. luminal A as the reference), even after adjusting for tumor size, LN status, radiotherapy, surgery type, chemotherapy, and hormone therapy. Garcia et al. [23] reported that patients with the luminal A subtype have the lowest prevalence of nodal involvement, as well as the lowest incidence of distant metastasis. Durbecq et al. [24] also reported that a significant proportion of patients aged >70 years develop luminal B-subtype tumors, which are associated with high proliferation, high grade, large size, and nodal invasion. We hypothesized that the luminal B subtype would involve higher grade tumors than would the other BCSs and accordingly compared the luminal A and B subtypes, which revealed significantly poorer RFS among elderly patients with the luminal B subtype and LN positivity. Lodi et al. [25] have reported that the differences in clinicopathological characteristics, increased incidence, and age-related mortality can be explained by biological changes in the breast, such as increased estrogen sensitivity, epithelial cell alterations, immune senescence, and tumor microenvironment modifications. However, these outcomes are also likely related to sociological factors, such as increased life expectancy, under-treatment, late diagnosis, and insufficient individual screening. The present study revealed similar results for survival outcomes, i.e., a significant difference in RFS according to the BCS, but only a non-significant difference in OS.

The present study has several potential limitations. For example, the study involved a retrospective analysis of data from a small sample of patients. However, to the best of our knowledge, this study included one of the largest series of patients reported in the literature to date. Thus, while previous studies have evaluated the association between LN status and individual BCSs, the prognostic value of the LN status and BCS has not been discussed for elderly breast cancer patients.

## Conclusions

In conclusion, the present study revealed differences in OS and RFS according to the BCS among elderly Taiwanese patients with breast cancer. The luminal B subtype was associated with especially poor RFS, and LN status was useful for predicting RFS in this subset of patients. The findings from our study may provide clinicians with more references for determining the prognosis and treatment strategies for elderly women with varying BCSs and LN statuses. In the future, research on the optimal clinical treatments for elderly women with different BCSs, LN statuses, and genetic profiles should be conducted, with particular focus on the types with poor OS and RFS.

## Author Contributions

**Conceptualization:** Kung-Hung Lin, Huan-Ming Hsu, Kuo-Feng Hsu, Chi-Hong Chu, Zhi-Jie Hong, Chun-Yu Fu, Yu-Ching Chou, Golshan Mehra, Ming-Shen Dai, Jyh-Cherng Yu, Guo-Shiou Liao.

**Data curation:** Chi-Hong Chu, Guo-Shiou Liao.

**Formal analysis:** Kung-Hung Lin, Huan-Ming Hsu, Kuo-Feng Hsu.

**Investigation:** Huan-Ming Hsu, Kuo-Feng Hsu.

**Methodology:** Kung-Hung Lin, Huan-Ming Hsu, Kuo-Feng Hsu, Zhi-Jie Hong, Golshan Mehra, Ming-Shen Dai.

**Project administration:** Huan-Ming Hsu, Zhi-Jie Hong, Ming-Shen Dai, Guo-Shiou Liao.

**Resources:** Huan-Ming Hsu, Kuo-Feng Hsu, Chi-Hong Chu.

**Software:** Kuo-Feng Hsu, Chi-Hong Chu.

**Supervision:** Huan-Ming Hsu, Chi-Hong Chu, Yu-Ching Chou, Ming-Shen Dai, Jyh-Cherng Yu, Guo-Shiou Liao.

**Validation:** Huan-Ming Hsu, Zhi-Jie Hong, Yu-Ching Chou, Guo-Shiou Liao.

**Visualization:** Huan-Ming Hsu, Yu-Ching Chou, Golshan Mehra, Jyh-Cherng Yu, Guo-Shiou Liao.

**Writing – original draft:** Kung-Hung Lin, Huan-Ming Hsu, Kuo-Feng Hsu, Chi-Hong Chu, Zhi-Jie Hong, Chun-Yu Fu, Yu-Ching Chou, Golshan Mehra, Ming-Shen Dai, Jyh-Cherng Yu, Guo-Shiou Liao.

**Writing – review & editing:** Kung-Hung Lin, Huan-Ming Hsu, Kuo-Feng Hsu, Chi-Hong Chu, Zhi-Jie Hong, Chun-Yu Fu, Yu-Ching Chou, Golshan Mehra, Ming-Shen Dai, Jyh-Cherng Yu, Guo-Shiou Liao.

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
