## [Decision Letter · Decision Letter 0]

25 Aug 2021

PONE-D-21-21337

Survival outcomes in elderly Taiwanese women according to breast cancer subtype and lymph node status: a single-center retrospective study

PLOS ONE

Dear Dr. Liao,

Thank you for submitting your manuscript to PLOS ONE. After careful consideration, we feel that it has merit but does not fully meet PLOS ONE’s publication criteria as it currently stands. Therefore, we invite you to submit a revised version of the manuscript that addresses the points raised during the review process.

ACADEMIC EDITOR: There are still some minor issues requiring to be addressed. Please kindly respond to the reviewers' comments. 

We look forward to receiving your revised manuscript.

Kind regards,

Jason Chia-Hsun Hsieh, M.D. Ph.D

Academic Editor

PLOS ONE

2. PLOS requires an ORCID iD for the corresponding author in Editorial Manager on papers submitted after December 6th, 2016. Please ensure that you have an ORCID iD and that it is validated in Editorial Manager. To do this, go to ‘Update my Information’ (in the upper left-hand corner of the main menu), and click on the Fetch/Validate link next to the ORCID field. This will take you to the ORCID site and allow you to create a new iD or authenticate a pre-existing iD in Editorial Manager. Please see the following video for instructions on linking an ORCID iD to your Editorial Manager account: https://www.youtube.com/watch?v=_xcclfuvtxQ.

Additional Editor Comments (if provided):

There are still some minor issues requiring to be addressed. Please kindly respond to the reviewers' comments.

Reviewers' comments:

Reviewer's Responses to Questions

**Comments to the Author**

1. Is the manuscript technically sound, and do the data support the conclusions?

Reviewer #1: Yes

Reviewer #2: Partly

2. Has the statistical analysis been performed appropriately and rigorously? 

Reviewer #1: I Don't Know

Reviewer #2: I Don't Know

3. Have the authors made all data underlying the findings in their manuscript fully available?

Reviewer #1: No

Reviewer #2: No

4. Is the manuscript presented in an intelligible fashion and written in standard English?

Reviewer #1: Yes

Reviewer #2: Yes

5. Review Comments to the Author

Reviewer #1: 1. What is the definition of the elderly in this study? The literature cited by the authors considers people 65 years of age or older as elderly.

2. How did the author screen out 554 cases? The inclusion and exclusion criteria should be specified in the Materials and Methods, and it is recommended to add a flow chart for subject screening.

3. The subjects in this study are patients from 2005 to 2015 and the data of these subjects are from electronic medical records. Has the electronic medical record been fully established for this period? If not, is there any way to make the data in this study as complete as possible? The authors should specify in the Materials and Methods.

4. Page 4, line 72: For "selected based on international recommendations", the authors should state clearly or cite relevant references.

5. Table 1: The total number of cases of all subtypes is 503, not 554.

6. Page 8, line 146-147: Some reference(s) should be cited for "many investigators have reported a statistically....and LN metastasis".

7. Page 8, line 149: According to the data in Table 3, "luminal HER2" should be corrected to "luminal B" here.

Reviewer #2: Question 1 - The use of person-years for incidence of breast cancer instead of an incidence rate in the Introduction is confusing. Also, a reference should be given for stat on the fastest-growing segment of the population.

It is unclear to me whether the use of immunohistochemistry used to identify ER/PR, HER2 status, etc. was done previously so the authors could access that data or if they performed the tests themselves on patient samples.

Question 3 - The authors state that the data cannot be shared publicly, but that it is available for certain researchers.

General comments - The numbers given for the number of HER2 positive patients as opposed to triple negative was surprising; there were more TN patients by several percentage points. Is this common in Taiwanese populations?

Table 1 comments - The tumor size was very large for 50% of the TN patients; it would be useful to know if this is because of late presentation. The numbers of modified radical mastectomies as opposed to breast conservation surgery are very high (which is surprising for elderly patients) but the least high for TN patients, unusual for the most aggressive form. Is this because the elderly patients did not want to undergo radiation after breast conserving surgery? It is not clear how much the patients' desires dictate treatment or what the standard of care is in Taiwan. There is no mention of specific HER2 therapy for HER2+ patients such as Herceptin or Trastuzumab; do the authors consider this chemotherapy and thus include it in the chemotherapy section or it is not available or not considered cost-effective for elderly patients in Taiwan? This needs to be clarified.

6. PLOS authors have the option to publish the peer review history of their article (what does this mean?). If published, this will include your full peer review and any attached files.

Reviewer #1: No

Reviewer #2: No

---

## [Author Response · Author response to Decision Letter 0]

1 Oct 2021

Dear Editor:

 Thank you for inviting us to submit a revised draft of our manuscript entitled “ PLOS ONE Decision: Revision required [PONE-D-21-21337]” on Aug 25, 2021. We also appreciate the time and effort you and each of the reviewers have dedicated to providing insightful feedback on ways to strengthen our paper. Thus, it is with great pleasure that we resubmit our article for further consideration. We have incorporated changes that reflect the detailed suggestions you have graciously provided. We also hope that our edits and the responses we provide below satisfactorily address all the issues and concerns you and the reviewers have noted. We revised the manuscript following the reviewers’ comments and carefully proofread the manuscript to minimize typographical, grammatical, and bibliographical errors. Here below is our description of revision according to the reviewers’ comments.

 To facilitate your review of our revisions, the following is a point-by-point response to the questions and comments delivered in your letter.

Part A 

Reviewer #1: 

1. The reviewer’s comment: 

What is the definition of the elderly in this study? The literature cited by the authors considers people 65 years of age or older as elderly.

The authors’ Answer:

In this study, the elderly is defined as 65 years of age or older. As the definition of old age is slightly different in different parts of the world, the current academic circles have no final conclusion on the specific age group of old age. The World Health Organization defines the elderly over 65 as the elderly. Despite growing research interest in the management of breast cancer in women over the age of 65, no internationally agreed recommendations exist for this population. The National Comprehensive Cancer Guidelines (NCCN) guidelines specifically define the elderly as patients over the age of 70 and recommend the use of elderly assessment tools for the elderly. The International Society of Geriatric Oncology (SIOG) guidelines define the elderly as patients over 65 years of age and provide evidence-based recommendations for the diagnosis and treatment of breast cancer in the elderly. In addition, if we study the definition of the elderly from 70 and 65 years old, the number of patients over 65 years old is indeed greater than the number of patients over 70 years old. In terms of studying various groups, it is easier to do research sample design

2. The reviewer’s comment:

How did the author screen out 503 cases? The inclusion and exclusion criteria should be specified in the Materials and Methods, and it is recommended to add a flow chart for subject screening.

The authors’ Answer:

Thank you for your examination and suggestion. We have added inclusion and exclusion criteria to Materials and Methods, with the following additions: According to the American Joint Cancer Council (AJCC) standards, the breast cancer database includes 4,363 newly diagnosed stage I, II, III, or IV patients in the Tri-Service General Hospital from 2005 to 2015. Exclude breast cancer patients< 65 years of age (n=3,809). The exclusion conditions are as follow patients with other cancer diagnoses before or after the initial breast cancer diagnosis, patients lacking correct ER, PR, HER2 data or missing data, diseases with unknown surgical information or missing surgical data after the initial breast cancer diagnosis, lack of relevant information patients with tumor size or the number of positive axillary lymph nodes and patients who have never had a disease or whose recurrence date is missing or wrong. Finally, a total of 503 patients were eligible for analysis between age and clinicopathological characteristics including 203 luminal A breast cancer, 128 luminal B breast cancer, 88 luminal B2 breast cancer, 20 HER2+ breast cancer, and 32 TN breast cancer (Fig. 1). 

LINE 110~122 PAGE 05~06

3. The reviewer’s comment:

The subjects in this study are patients from 2005 to 2015 and the data of these subjects are from electronic medical records. Has the electronic medical record been fully established for this period? If not, is there any way to make the data in this study as complete as possible? The authors should specify in the Materials and Methods.

The authors’ Answer:

We agree with you and have incorporated this suggestion throughout our paper. We add the following content: The hospital started to promote the use of electronic medical records in 2001. The Cancer Registration Center was formally established in 2002. Since then, the data of cancer patients began to be digitalized and registered year by year. At this time, electronic diseases are managed by using barcodes and radio frequency identification technology (radio frequency identification; RFID) to attach to the paper medical records, and then use database index and other technologies to record the location and flow of the paper medical records to achieve digitalization—the purpose of medical record management. From 2002 to 2009, our hospital adopted a mixed medical record model (paper medical records and electronic medical records exist at the same time). Because the legal status of electronic medical records has not been established and part of the electronic medical record system is not yet complete, our hospital adopts a transitional mixed medical record model. Between 2005 and 2009, we adopted large-scale manual methods to transfer paper medical records into electronic transcripts. At the same time, the construction of the electronic database of cancer patients in our hospital was completed from 2009 to 2010. Since then, the medical records of cancer patients have been all electronic. At the same time, it has also completed the goal of digitizing the paper medical records of all cancer patients in our hospital in the past. LINE 124~138 PAGE 06

4. The reviewer’s comment:

Page 4, line 72: For "selected based on international recommendations", the authors should state clearly or cite relevant references.

The authors’ Answer:

Thank you for your suggestion. The definition and classification of disease and treatment for breast cancer patients follow the international standards such as AJCC definitions, NCCN guidelines, and NICE guidelines. We have revised the cited references and attached the references. LINE 090~091 PAGE 04

5. The reviewer’s comment:

Table 1: The total number of cases of all subtypes is 503, not 554.

The authors’ Answer:

Thank you for your examination. We have revised the wrong part of the article. LINE 190 PAGE 08

6. The reviewer’s comment:

Page 8, line 146-147: Some reference(s) should be cited for "many investigators have reported a statistically....and LN metastasis".

The authors’ Answer:

Thank you for your examination. We have attached relevant references

LINE 216~217 PAGE 11

Edge SB, Compton CC. The American Joint Committee on Cancer: the 7th edition of the AJCC cancer staging manual and the future of TNM. Annals of surgical oncology. 2010;17:1471-1474.

Badwe R, Hawaldar R, Nair N, et al. Locoregional treatment versus no treatment of the primary tumour in metastatic breast cancer: an open-label randomised controlled trial. The Lancet. Oncology. 2015;16:1380-1388.

Graham LJ, Shupe MP, Schneble EJ, et al. Current approaches and challenges in monitoring treatment responses in breast cancer. Journal of Cancer. 2014;5:58-68.

7. The reviewer’s comment:

Page 8, line 149: According to the data in Table 3, "luminal HER2" should be corrected to "luminal B" here.

The authors’ Answer:

Thank you for your examination. We have revised the wrong part of the article. LINE 219 PAGE 11

Part B 

Reviewer #2: 

1. The reviewer’s comment: 

The use of person-years for incidence of breast cancer instead of an incidence rate in the Introduction is confusing. Also, a reference should be given for stat on the fastest-growing segment of the population. It is unclear to me whether the use of immunohistochemistry used to identify ER/PR, HER2 status, etc. was done previously so the authors could access that data or if they performed the tests themselves on patient samples. 

The authors’ Answer: 

Thanks for your comment. We rewrote relevant and difficult sentences. The rewritten paragraph is as follows: Breast cancer is the most common cancer among women and the leading cause of cancer-related deaths worldwide [2]. The Taiwanese National Health Insurance database was used to estimate the annual prevalence and incidence of breast cancer between 1997 and 2013, which revealed a prevalence of 834.37 per 100,000 persons and an incidence of 93.00 per 100,000 persons in 2013. In Taiwan, the standardized incidence was 52.34 per 100,000 person-year in 1997 and 93.00 per 100,000 person-year in 2013 [1,3]. The age-standardized incidence rates (ASIR) have gradually increased over the past several years, with an incremental annual change of 3.5 per 100,000 persons. It suggests that the breast cancer incidence increased over the study period [2,3]. In addition, we have previously tested patient samples and used immunohistochemistry to identify ER/PR, HER2 status, etc., to study the data. So this time, we will not repeat it.

LANE41~50 PAGE 3

2. The reviewer’s comment:

The numbers given for the number of HER2 positive patients as opposed to triple negative was surprising; there were more TN patients by several percentage points. Is this common in Taiwanese populations?

The authors’ Answer: 

To avoid readers' misunderstanding, we have changed Luminal HER2 to Luminal B2 to make it easier to understand. The distribution of BCSs among the 554 elderly Taiwanese women (>65 years old) was as follows: luminal A (46.7% of patients), luminal B (25.4% of patients), Luminal B2 (17.5% of patients), HER2 (4.0% of patients), and TN (6.4% of patients). There are indeed slightly more patients in the TN group than in the HER2 group. The reason is that some patients lost contact or refused to cooperate during the long-term follow-up of the patients and withdrew from the study. We have deducted the in complete part of the data, so the data numbers make the TN group look slightly more significant. However, the two groups of Luminal B2 and HER2 patients together account for approximately 21.5%. According to research published in the past, the proportions of luminal-A, luminal-B, HER2-positive, and triple-negative subtype are 50%~60%, 15%~20%, 15~20%, 8%~37%, respectively. Therefore, if we add up the number of Luminal B2 patients and the number of HER2 patients in our study, 108 (21.5%) are in line with the digital content reported in the previously published papers. From the data in the past few years, the number of HER2-positive patients is indeed on the rise. In the past few years, there were more than 10,000 new patients in Taiwan each year, of which a quarter was HER2-positive. It is estimated that there will be an increase of 3,000 HER2-positive breast cancer patients in Taiwan each year. In addition, a 2009 study by Lin CH et al. pointed out that the prevalence of breast cancer in Taiwanese elderly (>50 years old) breast cancer patients are higher than that of young people (≤50 years old) breast cancer patients with basal-like subtypes (17% vs. 9%)

3. The reviewer’s comment: 

Question 3-Table 1 comments - The tumor size was very large for 50% of the TN patients; it would be useful to know if this is because of late presentation. The numbers of modified radical mastectomies as opposed to breast conservation surgery are very high (which is surprising for elderly patients) but the least high for TN patients, unusual for the most aggressive form. Is this because the elderly patients did not want to undergo radiation after breast conserving surgery? It is not clear how much the patients' desires dictate treatment or what the standard of care is in Taiwan. There is no mention of specific HER2 therapy for HER2+ patients such as Herceptin or Trastuzumab; do the authors consider this chemotherapy and thus include it in the chemotherapy section or it is not available or not considered cost-effective for elderly patients in Taiwan? This needs to be clarified.

The authors’ Answer:

I. The tumor size for the TN patients of breast cancer in this study is indeed relatively large, but this is the result of our research and statistical analysis. The reason is speculated that it may be related to the lack of screening concepts of patients in the past, the insufficient popularization of suitable screening equipment, and patients' delay in seeking medical treatment.

II. In this study, the proportion of TN patients undergoing MRM surgery is relatively large. We speculate that the reason may be that if elderly breast cancer patients decide to receive treatment, the patient pays more attention to their own health than breast beauty and breast functionality. As far as the local people in Taiwan are concerned, elderly breast cancer patients believe that they will be healthier and less likely to relapse after undergoing MRM surgery. It is not because elderly patients do not want to receive radiotherapy after breast-conserving surgery.

III. Our standards for treating breast cancer patients are based on our hospital’s breast cancer treatment guideline, which is modified majority by the direction of the National Comprehensive Cancer Network (NCCN) guidelines and the National Institute for Health and Clinical Excellence (NICE) guidelines. For the current hormone therapy for specific HER2 therapy for HER2+ patients, we will first assess whether the patient's heart's left ventricular ejection fraction (LVEF) is normal, and the patient's LN positive can be administered. For elderly breast cancer patients in Taiwan, Herceptin is available and considered cost-effective. However, the cost of Herceptin drugs is too expensive for patients. Herceptin was not officially included in Taiwan's National Health Insurance until January 1, 2010. Prior to this, patients with breast cancer in Taiwan must use Herceptin at their own expense. Most patients cannot afford expensive drugs at their own expense, so most patients gave up Herceptin treatment before 2010. In addition, targeted therapy (Herceptin or Trastuzumab) generally requires combined chemotherapy to treat breast cancer effectively. Here, the chemotherapy used is paclitaxel (PTX). In addition, we will arrange surgery for patients with early TN breast cancer. On the contrary, we will not arrange surgery for patients with metastatic TN breast cancer.

IV. According to the study of Kuo CN et al. in 2020, breast cancer treatment in Taiwan is currently as follows: The treatment of early-stage breast cancer is comprehensively determined by tumor size, nodal status, and expression of ER, PR, and human epidermal growth factor receptor-2 (HER2). For surgically resected hormone-receptor-positive breast cancer, adjuvant hormonal therapy is always indicated. Five-year treatment of tamoxifen plus exemestane or 5-year tamoxifen following 5-year letrozole are options for postmenopausal women, while ovarian suppression therapy plus tamoxifen is an alternative to chemotherapy for the selected premenopausal woman. In HER2-positive patients, one-year adjuvant trastuzumab has been approved for the node-positive disease. In addition to anthracycline, paclitaxel or docetaxel is generally the primary therapy offered to patients with node-positive cancer. In patients with metastatic disease, tamoxifen, letrozole, anastrozole, and exemestane are the treatment of choice for hormone-positive breast cancer. For letrozole-failed hormone-receptor-positive disease, everolimus plus exemestane is a viable treatment option. (After Dec. 2019, CDK46i was approved by our national health insurance for HR+ mBC). Trastuzumab-based treatment is an option for breast cancer with overexpression of HER2.

CONCLUDING REMARKS: 

Again, thank you for giving us the opportunity to strengthen our manuscript with your valuable comments and queries. We have worked hard to incorporate your feedback and hope that these revisions persuade you to accept our submission. Many grammatical or typographical errors have been revised. All the lines and pages indicated above are in the revised manuscript. We acknowledge the reviewer’s comments and suggestions very much, which are valuable in improving the quality of our manuscript. Thank you and all the reviewers for the kind advice.

Sincerely yours,

Guo-Shiou Liao, MD.

Corresponding Author

Division of General Surgery, Department of Surgery, 

Tri-Services General Hospital, National Defense Medical Center, 

No 325, Sec. 2, Cheng-Kung Rd, Nei-Hu 114, Taipei, Taiwan, R.O.C.

Tel: +886-2-87927191;Fax: +886-2-87927273

E-mail: guoshiou@ndmctsgh.edu.tw

---

## [Editor Report · Decision Letter 1]

29 Nov 2021

Survival outcomes in elderly Taiwanese women according to breast cancer subtype and lymph node status: a single-center retrospective study

PONE-D-21-21337R1

Dear Dr. Liao,

We’re pleased to inform you that your manuscript has been judged scientifically suitable for publication and will be formally accepted for publication once it meets all outstanding technical requirements.

Kind regards,

Jason Chia-Hsun Hsieh, M.D. Ph.D

Academic Editor

PLOS ONE

Additional Editor Comments (optional):

All the issues were answered adequately.
---

## [Editor Report · Acceptance letter]

13 Dec 2021

PONE-D-21-21337R1 

Survival outcomes in elderly Taiwanese women according to breast cancer subtype and lymph node status: a single-center retrospective study 

Dear Dr. Liao:

I'm pleased to inform you that your manuscript has been deemed suitable for publication in PLOS ONE. Congratulations! Your manuscript is now with our production department. 

Kind regards, 

on behalf of

Dr. Jason Chia-Hsun Hsieh 

Academic Editor

PLOS ONE